# What makes a city breastfeeding friendly? A qualitative analysis of interviews with breastfeeding women from Europe and Asia

**May Loong Tan**[1,2]*, **Elizabeth J. O'Sullivan**[3], **Jacqueline J. Ho**[1], **Amal Omer-Salim**[4], **Fionnuala M. McAuliffe**[2]

**1** Department of Paediatrics, RCSI & UCD Malaysia Campus, Penang, Malaysia, **2** UCD Perinatal Research Centre, School of Medicine, National Maternity Hospital, University College Dublin, Dublin, Ireland, **3** School of Biological, Health and Sports Sciences, Technological University Dublin, Dublin, Ireland, **4** World Alliance for Breastfeeding Action, Penang, Malaysia

* may.tan@ucdconnect.ie, mltan@rcsiucd.edu.my

## Abstract

### Background

The warm chain of support is the continuous enabling environment from the mother's first contact with healthcare professionals during early pregnancy, birth and immediate post-partum period, her transition from healthcare facility to home, through to work and the community at large. A breastfeeding-friendly city should be able to support a breastfeeding journey across the warm chain.

### Objective

To determine breastfeeding women's perspective of an ideal breastfeeding-friendly city.

### Methods

Between September 2021 and January 2022, twenty-two women who were breastfeeding or had ever breastfed in the last 5 years from Ireland and Malaysia were interviewed. A set of selection criteria was applied to ensure representation of a range of the characteristics known to be associated with breastfeeding success: diverse age groups, birth and breastfeeding experiences, culture and socioeconomical background. One-on-one semi-structured online interviews were conducted by the first author. Data were analysed using Braun and Clarke's Thematic Analysis framework.

### Results

One overarching theme of breastfeeding at the front and centre of the city, and three major themes were developed: 1. mothers feel supported when breastfeeding is prioritised; 2. when breastfeeding is visible in the environment, it becomes normalized; 3. there is a need to have seamless breastfeeding support across the continuum of the warm chain, and at all levels of society.

**Data Availability Statement:** This manuscript's data cannot be made publicly available as the data in this paper comprises confident transcripts of

interviews. These restrictions have been imposed by the Joint Penang Independent Ethics Committee and UCD Research Ethics Committee. However, interested parties may request access to the data via email to the UCD Academic Administrator (stephanie.begley@ucd.ie). Data access may be granted on reasonable request to eligible parties and with the completion of any required prerequisites (such as a Data Use Agreement).

**Funding:** The author(s) received no specific funding for this work.

**Competing interests:** The authors have declared that no competing interests exist.

## Conclusions

The findings demonstrated the importance of prioritized, and continuous support throughout the breastfeeding journey. The hopes and aspirations of a breastfeeding-friendly city expressed here would be useful for cities to consider when developing or implementing breastfeeding support programmes as well as guide development of indicators of a breast-feeding-friendly city.

## Introduction

Breastfeeding is essential for the health and well-being of the newborn and the mother, with short- and long-term benefits for both [1–3]. Beyond health, the economic benefits of a community with high breastfeeding rates are also significant [4]. The optimal duration of breastfeeding is exclusive for the first 6 months and continued until 2 years and beyond [5]. Despite these, not every mother who has the intention to breastfeed will succeed [6, 7]. Breastfeeding is not always easy, and multiple factors contribute to the early cessation of breastfeeding. Reasons such as latching difficulties, pain, perceived milk insufficiency, and work were universally reported by mothers, including those in Ireland and Malaysia, as reasons to stop breastfeeding before the recommended duration of at least 2 years [8–10]. There are also evidence showing how non-breastfeeding-friendly practices, such as early supplementation of formula in the hospital, can shorten the duration of breastfeeding despite good maternal intentions [11]. Apart from these, factors such as wellbeing, self-efficacy, adequate sleep, supportive partner, and healthcare professionals, also contributes to breastfeeding outcomes [2, 12–14]. Seeing that internal and external factors contribute to success of breastfeeding, it is imperative that an enabling environment throughout the breastfeeding journey is in place for each woman and her family.

An enabling environment for breastfeeding starts from the woman's first contact with healthcare professionals during early pregnancy. It continues at birth and immediate post-partum period, and subsequently her transition from the healthcare facility to home, work, and the community. The community includes public spaces such as parks, public transport as well as other 'open' places a breastfeeding woman might visit such as markets, shops, restaurants, and places of worship. The enabling environment could also precede these present-day environments, such as early exposure to breastfeeding as a child and education on breastfeeding in school. The continuation and connectivity of these diverse supports through the duration of breastfeeding have been referred to as the warm chain of support for breastfeeding [15]. The warm chain is akin to the cold chain in vaccine delivery whereby a break within the chain would compromise the outcome, in this case, the duration of breastfeeding. Within the warm chain are 'actors', who are people that play a role in extending support to the mother-infant dyad at a specific point of time. See Fig 1 for a diagrammatic representation of the warm chain.

### The conceptual models of determinants of breastfeeding

One of the earliest conceptual models of determinants of breastfeeding was the ecological breastfeeding model described by Hector *et al.* [16]. In this framework, the authors described breastfeeding as being influenced by individual level, group level and society level factors. Individual level factors include intentions, past experience, birth experience, and health. Group level factors relate to the environment within which the mother and baby reside in such as

| Mother and baby timeline | Pregnancy | | Child's 2nd birthday (and beyond) |
|---|---|---|---|

| Initiatives | Antenatal care | Delivery and Postpartum Care | Care beyond 6 weeks |
|---|---|---|---|

| Actors | Dietitians/Nutritionists | Healthcare providers | Peer support | Trade union |
|---|---|---|---|---|
| | Family members | Lactation consultants | | Employers/Employees |

| Setting | Healthcare | Community | Workplace |
|---|---|---|---|

**Fig 1. Diagrammatic representation of the warm chain of support for breastfeeding (adapted with permission from World Alliance for Breastfeeding Action (WABA).**

healthcare, community, and workplace. The society level factors referred to cultural norms related to breastfeeding, parenting, and women in general.

A more recent model of the determinants of breastfeeding was described by the authors of the Lancet Breastfeeding Series 1 [17]. In this framework, three levels of determinants influence a mother's breastfeeding behaviour–structural, settings, and individual. The structural determinants are the fundamental background influences of social and culture, shaped not only by cultural beliefs but also by marketing of infant formula. The settings described in this model were healthcare, community, and workplace. The individual level relates to the mother-infant dyad as well as their relationship with other people around them. The warm chain concept is a step up from these models, with the emphasis on the connection between all the different components of the determinants of breastfeeding. A breastfeeding-friendly city is an example of how elements of these different models are operationalized simultaneously.

In line with a call to scale up breastfeeding support [18], cities around the world have become "breastfeeding-friendly" or declared themselves to be such [19–21]. Despite the recognized need for supportive environments, there is currently a lack of consensus on what constitutes a breastfeeding-friendly city, hindering the development of effective interventions and policies. Our scoping review found that different criteria and definitions have been used for each city and there are no consistent and specific indicators of a breastfeeding-friendly city are presently available [22]. Therefore, our research team set out to develop a novel set of indicators of a breastfeeding-friendly city. Generally, indicators for breastfeeding support programmes or interventions have not consistently included the perspectives of breastfeeding

women. This included the early versions of the Baby Friendly Hospital Initiative [23]. This is not surprising as Public-Patient-Involvement or co-production of research was not common practice until the last few years [24]. As breastfeeding women are 'users' of breastfeeding friendly cities, they should be consulted to find out what matters to them. Policies and interventions developed without input from end-users may fail to gain widespread adoption, as they may not adequately address their needs. Similarly, a breastfeeding-friendly city must be designed with the active participation of breastfeeding women.

This study aims to explore the perspectives of breastfeeding women from two continents–Europe and Asia, to understand the key elements of an ideal breastfeeding-friendly city. A list of items obtained from the interviews was used in a Delphi study as part of the development of a new indicator set of a breastfeeding-friendly city [25]. This report describes the detailed, inductive analysis of the interviews.

## Methods

To achieve our objective, we conducted one-on-one semi-structured online interviews with a diverse group of women from Ireland in Europe and Malaysia in Asia.

Women were eligible to participate if they were over 18 years old and had ever breastfed a baby in the last 5 years. "Ever breastfed" was defined according to WHO Indicators for Assessing Infant and Young Child Feeding as anyone who had provided her own or another baby (i.e., adopted baby or wet nursing) any amount of breastmilk directly from her breasts or as expressed breastmilk [26]. Participants were ineligible if they were unable to converse in English or Malay or had no internet access for the interview. The study was conducted following the principles of the Declaration of Helsinki and received approval from Joint Penang Independent Ethics Committee (No. JPEC 21–0011, 4 July 2022), as well as declared as review exempt from UCD Human Research Ethics Committee (No: LS-E-22-139-Tan-McAuliffe, 14 July 2022). Electronic written consent was obtained before the commencement of the interviews, which included consent for the recording of the interviews and use of pseudonyms when quoting them.

### The setting

This study was conducted with participants from Ireland and Malaysia. Ireland is a European, high-income country with a low breastfeeding rate (31.1% exclusive breastfeeding at 3 months, 2021) [27] while Malaysia is a Southeast Asian, upper-middle income country with a modestly higher breastfeeding rate (47.1% prevalence of exclusive breastfeeding under 6 months, 2016) [28]. Although both countries have a national breastfeeding policy in place, the differences in the cultural and economic background of the two countries could have influenced the practice of breastfeeding. Therefore, including participants from these countries provided a global perspective to the subject as well as an opportunity to compare differences in views.

### Sampling method and sample size

Information about the study was circulated to two online (Facebook) breastfeeding support groups, one in Ireland and the other in Malaysia. Prior to posting the information about the study on the groups' Facebook pages, the administrators of the groups were contacted for approval. Collectively, the two breastfeeding groups had over 120,000 members. In the post on Facebook, we asked eligible and interested women to contact the primary investigator. Women who responded to the call to participate were asked to complete a screening questionnaire to identify a sample that was diverse based on the characteristics below (listed in

alphabetical order). The characteristics were selected based on known individual level factors associated with breastfeeding [29, 30].

1. Age

2. Breastfeeding experience (first time, had breastfed more than 1 child)

3. Education background (highest educational attainment)

4. Employment status (full time employment, stay-at-home mother)

5. Ethnic and cultural background

6. Had a preterm baby and/or twins

7. How they breastfeed (direct breastfeeding, expressed breast milk feeding)

8. Weight and height for calculation of body mass index

To obtain the perspectives of a wide range of women who differed on characteristics known to be associated with breastfeeding, we used purposive sampling. We reviewed the characteristics of the participants and selected them for the interviews based on the characteristics. The decision for selection of a participant was made by the lead investigator in consultation with the other investigators. As the interviews went on, personal contacts of the investigators were also used to source participants with the predetermined characteristics that were yet represented in the study.

The sample size was determined by two factors: ethnographic and information saturation. We recruited participants until all categories of the characteristics of women listed above were represented. The interviews also continued until the point where no new information about what makes a city breastfeeding-friendly was forthcoming.

## Demographic data collection

Women who expressed an interest and consented to be interviewed, completed an online screening form before the interview. The screening form collected demographic data such as age (categorical), ethnic background, education level, employment status, method of breastfeeding (direct, expressed or combination of both), past breastfeeding experience, self-reported weight, self-reported height, and details about their baby (prematurity (less than 37 weeks gestation), twins or admission to neonatal intensive care). The women were not given any renumeration for their participation.

## Selection of participants for interview

All the women who completed the online and screening form, and met the inclusion criteria, were contacted for interviews. Initially, the women were contacted based on the timing of receiving the forms. They were contacted within one week of receiving the screening form, and if they responded, interviews scheduled within two weeks. If a participant did not respond to our contact for an interview appointment, the next person with similar characteristics was selected. As the number of participants grew, preference was then given to those with characteristics not yet met in the earlier participants. This continued until information saturation was met.

## The interviews

The interviews were conducted by a single interviewer (MLT from Malaysia) *via* a video conferencing platform (Zoom). One participant from Malaysia was interviewed over the phone

because she did not know how to use a video conferencing application despite having internet access. This was reported back to the ethics committee at the end of the study.

The interviews were conducted either in English or Malay, according to the participants' preference and were recorded (video and audio) using the record function of the video conferencing platform. MLT is fluent in both languages. The sole telephone interview was recorded using an external audio recorder. The interviews were structured to ask about the supportive or enabling factors for breastfeeding at healthcare facilities, workplaces (if working) and public areas, negative breastfeeding experiences at healthcare facilities, workplaces (if working) and public areas, and expectations of an ideal breastfeeding-friendly city. The interview questions were pre-tested in a separate group of breastfeeding women from Ireland and Malaysia prior to the study (3 Malaysians, 1 Irish). The pre-testing was to ensure that the questions were clear in both the English and Malay languages. After the pre-testing, minor modifications were made to the questions, and additional probes were included. For example, we added questions about transitioning from hospital to home and support during the first 6 weeks. See Box 1 for the questions and probes.

### Box 1. Questions and probes used in the semi-structured interviews.

I am doing a study to develop markers/indicators of a breastfeeding friendly city. Because a mother moves through different sectors that exist in a city, I would like to hear your experiences relating to breastfeeding in various places.

Before we start, could you please tell me a bit about yourself and your breastfeeding journey thus far?

Let's start with your experiences in the hospital setting. What do you recall that was helpful to you? What was not helpful?

What about your experience transitioning from hospital to home?

What about support at home in the first 6 weeks?

Next, let's talk about work. In your work, what was helpful? What was not?

What about transitioning from home to work?

We now move to the last setting–public places, these can include shops, restaurants, churches etc. What was helpful? What was not?

Let's say your city is now declared breastfeeding friendly. What does it look like to you?

What do you think we should do to achieve that?

Do you understand what is breastfeeding friendly?

### Data management and analysis

All interview recordings were transcribed verbatim in the original language used. All transcriptions were pseudo-anonymised before translations and analysis. Transcriptions of

interviews conducted in Malay were translated into English by MLT and were cross-checked by an independent person (a colleague of MLT who is fluent in both languages). To avoid identification, we used pseudonyms when quoting the participants.

We used NVivo 13 to organize and manage the data analysis [31]. The qualitative analysis was conducted using reflexive thematic analysis as described by Braun & Clarke [32]. After the initial reading of the transcripts, the data were grouped into the pre-specified sectors of a city (hospital, workplace, community, and public spaces). This also helped with the familiarization with the data. Next, data were coded in two phases: first, within each sector and subsequently across the entire dataset irrespective of sector. The coding process used both semantic (mainly for codes within a sector) and latent (mainly for codes across the dataset) codes. MLT and EJOS independently read the interview transcripts. MLT conducted the grouping and coding as well generating the initial themes. MLT and EJOS then reviewed the initial themes together and defined them further. Both MLT and EJOS (from Ireland) are active researchers and advocates for breastfeeding and have constant interactions with breastfeeding women in their professional capacities. They had also breastfed their own children.

Throughout the process of defining the themes, we used an inductive approach, i.e. observation (what the mother reports), pattern recognition (what mothers report in all sectors) and theory development (final themes). The themes were presented to the remaining authors (JJH, AOS and FMM) to ensure objectivity and address any bias that may have occurred. All authors reviewed and agreed on the final themes.

## Results

Interviews were conducted between September 2021 and January 2022 with women from Malaysia and Ireland. During this period, both countries were still in partial lockdown due to the COVID-19 pandemic and many of the women had their babies born during the full lockdown period of the pandemic. Forty-two women expressed interest in participating in the study. We interviewed a total of 22 women from a variety of ethnic/cultural backgrounds living in Malaysia (n = 11) and Ireland (n = 11). The equal number in the two countries was coincidental. Four women did not respond to our invitation for an interview and 16 were not needed after selection based on the demographic characteristics and subsequent information saturation. Eighteen interviews were conducted in English and four in Malay language. The interviews lasted an average of 16:24 minutes (range 9:30 to 20:30 minutes). The shorter interviews involved participants who were less forthcoming and needed a lot of prompting. All the predetermined characteristics were present among the participants. The participants from Ireland were generally older with higher education level compared to the Malaysians. Table 1 shows a summary of the characteristics of participants.

### Overarching theme: Breastfeeding front and centre

From the interviews, we developed one overarching theme and three major themes. A summary of the overall findings is presented in Table 2.

The overarching theme was that a breastfeeding-friendly city was one where breastfeeding is at the front and centre of each component of the warm chain. For a city to be truly breastfeeding-friendly, breastfeeding should be brought out to the forefront through prioritization and visualization of breastfeeding, as well as ensuring that support for breastfeeding is continuous across the warm chain. The women recognised that to achieve this, there must be clear leadership, thoughtful facilities, and a welcoming culture.

As Dinah (Malaysia) puts it, "It is not possible for a town to be breastfeeding-friendly if it didn't come from a leader." With clear leadership, any activities related to breastfeeding will

**Table 1. Demographic and characteristics of participants.**

| Characteristics | Ireland (n = 11) | Malaysia (n = 11) | Total (n = 22) |
|---|---|---|---|
| Age | | | |
| Between 21–30 years old | 2 | 5 | 7 |
| Between 31–40 years old | 9 | 5 | 14 |
| Above 40 years old | 0 | 1 | 1 |
| Ethnicity/Cultural background | | | |
| White (Irish) | 9 | 0 | 9 |
| White (Mexican) | 1 | 0 | 1 |
| Asian (Malay) | 0 | 4 | 4 |
| Asian (Chinese) | 1 | 4 | 5 |
| Asian (Indian) | 0 | 1 | 1 |
| Asian (Filipino) | 0 | 1 | 1 |
| Body Mass Index | | | |
| Less than 24.9 kg/m$^2$ | 6 | 7 | 13 |
| Between 25 to 29.9 kg/m$^2$ | 3 | 3 | 6 |
| Equal or above 30 kg/m$^2$ | 2 | 1 | 3 |
| Highest Education attainment | | | |
| Certificate or Diploma | 0 | 3 | 3 |
| Bachelor's Degree | 3 | 7 | 10 |
| Higher degree (Masters or PhD) | 8 | 1 | 9 |
| Employment status | | | |
| Full-time | 10 | 4 | 14 |
| Part-time | 1 | 2 | 3 |
| Self-employed | 0 | 1 | 1 |
| Stay-at-home mother | 0 | 4 | 4 |
| Neonatal complications [a] | | | |
| Twins | 0 | 2 | 2 |
| Preterm delivery | 3 | 2 | 5 |
| Baby in NICU | 3 | 3 | 6 |
| Past breastfeeding experience | | | |
| First-time breastfeeding | 4 | 2 | 6 |
| Breastfed 2 children | 6 | 2 | 8 |
| Breastfed 3 or more children | 1 | 7 | 8 |
| Method of breastfeeding | | | |
| Only direct feeding | 7 | 2 | 9 |
| Mixed direct and expressed feeding | 4 | 9 | 13 |
| Only expressed breast milk feeding [b] | 1 | 0 | 1 |

Footnotes:

[a] One baby may have more than one complication.

[b] One of 3 children.

have direction and be in a position of importance. She described how in her area, where she volunteers in a breastfeeding support group, they will always invite a local leader to attend any event organized by the group. The support demonstrated by their presence at these events may be small but sends an important message towards making a city breastfeeding friendly. Priority should also be seen when considering facilities for breastfeeding, where is "not on the second floor all the time" *(Olivia, Malaysia)*. A welcoming culture among people in the city also brings

**Table 2. Themes and subthemes.**

| Theme | Description of theme | Subthemes |
|---|---|---|
| Overarching theme: Breastfeeding at the front and centre of every city. | Overall perspective of a city where breastfeeding is brought to the forefront and how it can be achieved. | |
| Major Theme 1: Mothers feel supported when breastfeeding is prioritised. | Areas of support, and where they were helpful or otherwise. | *Highest priority noted during the perinatal period*: There was no shortage of support offered to healthy mothers and their newborns during this period. |
| | | *Breastfeeding becomes secondary when the baby or mother is sick.* |
| | | *Sometimes breastfeeding support felt like an after-thought.* This is most evident beyond the perinatal period. |
| Major Theme 2: When breastfeeding is visible in the environment, it becomes normalized. | Normalization of breastfeeding in public. | *Effects of breastfeeding visibility*: Examples of how seeing other women breastfeeding made an impact on decision to breastfeed. |
| | | *Visuals of breastfeeding as part of everyday life*: A description of visuals of breastfeeding in media (print and electronic). |
| | | *Differing views about breastfeeding in public*: Not everyone views public breastfeeding with similar enthusiasm. |
| | | *Public breastfeeding rooms are for the babies' comfort*: The mothers are not particularly concerned about modesty. |
| Major Theme 3: There needs to be seamless breastfeeding support across the continuum of the warm chain, and all levels of society. | Focused on the warm chain across sectors and equity of support. | *Discontinuity of support across sectors*: There is no continuity of support when mothers move from one sector to another. |
| | | *Closing the gaps within the sector*: Within each sector there are areas of support that could be addressed better. |
| | | *Inequitable access to breastfeeding support*: Not everyone could afford breastfeeding support. |

breastfeeding to the fore front. However, this welcoming culture is not always experienced by the women. There needs to be a culture change away from bottle feeding if a city is to be breastfeeding friendly.

> "Our culture is so geared to maybe more bottle feeding and not breastfeeding in public, I think we need those things (stickers that say 'Breastfeeding welcomed here") to help it along until it becomes more normalized." *Jade, Ireland*

Camelia (Ireland) puts it aptly when talking about what such a breastfeeding-friendly city looks like to her, ". . . and change the people living there!"

## Major Theme 1: Mothers feel supported when breastfeeding is prioritised

This theme describes situations in which the mothers felt most supported during their breast-feeding journey. These situations were related to the amount of priority placed on breastfeeding. The subthemes are highest priority during perinatal period especially for healthy babies, breastfeeding becomes secondary when the baby or mother is sick, and beyond the perinatal period, some breastfeeding support felt like an after-thought.

**Highest priority for breastfeeding noted during the perinatal period.** There was no shortage of the types of support provided for breastfeeding during the perinatal period. The mothers spoke about antenatal breastfeeding classes, being given skin-to-skin immediately after birth and help with breastfeeding while in the hospital after delivery. After discharge

from hospital, many women had a support group to go to for help to continue breastfeeding. These supports were mainly directed at making sure breastfeeding gets off to a good start.

"I did the prenatal breastfeeding course, you know, that's offered, so that was good as a starting point" *Eva, Ireland*

"The first golden hour they didn't take the baby away for cleaning or whatsoever, and the baby got to bond with me, and she can immediately try to latch" *Melanie, Malaysia*

"The nurse sort of like stayed on the first time they bring her to breastfeed....she really sat, like, the whole hour (and it) was actually really helpful" *Olivia, Malaysia*

"My local health centre had a weekly breastfeeding group with the public health nurse, and that was great" Grac*e, Ireland*

For those who did not have or could not access a physical in-person group, the mothers reported that support from online breastfeeding groups gave them the help and encouragement to continue breastfeeding.

"I almost quit, and I found a Facebook group for moms and I just posted there, and someone told me about breastfeeding specific group and I joined that and that saved my journey" *Isabelle, Ireland*

"Then after that, I joined [name of Facebook group]. And it was helpful because of the community support, even though you don't get it face to face." *Elise, Malaysia*

Breastfeeding support groups provided not only specific support for breastfeeding but also parenting and social support. In addition, it gave the mothers an opportunity to offer support to others who had similar experiences.

"For me, [*name of group*] was a lifeline because I've made all of my best friends through (this group)" *Nikki, Ireland*

"And then once you know the answer, you can also offer support (*to others*)" *Elise, Malaysia*

**Breastfeeding becomes secondary when the baby or mother is sick.** Unlike the healthy babies, babies that required special care, breastfeeding seemed to take second place in the eyes of healthcare professionals, which some mothers expressed disappointment about. The mothers could understand the need to prioritize the medical problem but did not appreciate that formula feeding seemed to be presented by healthcare professionals as the more appropriate way of feeding. Upon reflection, when babies missed the opportunity to exclusively breastfeed, the mothers expressed disappointment. It appears they wished someone gave breastfeeding a chance.

"So [they] would come around, they will ask you questions like, you know, indirectly they're asking you, why don't you top up with formula milk, since his reading (bilirubin) is very high" *Stella, Malaysia; narrating the incidence of her baby admitted for neonatal jaundice.*

"But I'm kind of annoyed that (formula) was the first suggestion, instead of let's get more colostrum and syringe feed him colostrum." *Lilia, Ireland; in response to being told that her baby in NICU needed formula.*

Mothers who are unwell after delivery are sometimes separated from their newborns. In hospitals, location of the wards for mothers and babies should consider breastfeeding. Although it was not viewed as an inconvenience, Lilian (Ireland) described how she was "going up and down every time for feed". She was admitted post-delivery due to a medical issue and her baby required phototherapy for jaundice. Another mother found it difficult to go to the hospital breastfeeding room to feed her twin preterm babies.

"..the nursing room is quite far, maybe 2–3 minutes' walk" *Paula, Malaysia*

This distance, though it may not be considered far by some, proved to be difficult for her as she was post-caesarean section and had twins. She felt alone as well because her husband could not help her as the room was reserved for mothers only.

These examples demonstrated that while there was a lot of support put in place, they were prioritized for the healthy newborn and for the babies that needed hospitalization, or mothers who needed extra care, breastfeeding didn't come across as being a priority.

**Sometimes breastfeeding support felt like an after-thought.** The experiences shared by the mothers of breastfeeding beyond the perinatal period and the hospital setting highlighted the inadequacy of how some supports were delivered. In many public places, breastfeeding rooms were set up for mothers to use. However, some of these rooms were so inconvenient that the mothers felt that the people responsible for setting them up were merely fulfilling a checklist. For example, a breastfeeding room that is locked, and the key is not easily available or a room located on the top floors where it is more difficult to get to with a baby in a stroller.

"The breastfeeding room was locked. I had to get the key from the guard but it was too slow" *Ruby, Malaysia*. (She ended up breastfeeding her baby on the chair in the open)

"I feel like our nursing rooms should be on every floor of whatever malls or supermarkets" *Olivia, Malaysia*

These examples highlighted that the people who provided the breastfeeding rooms may had not thought about the usability of the rooms or understood how breastfeeding works.

In the workplace, many women faced the similar problems and described situations where breastfeeding was an afterthought. On one hand, many reported that their workplace did not stop them from breastfeeding or expressing breast milk during work but on the other; however, they highlighted that the support provided was not well thought through. The primary example of this was the provision of breastfeeding rooms at workplaces. Unfortunately, participants reported that some employers still considered the toilet to be a suitable private room to offer to their employees.

"Then, when I started work at six months. . . (my workplace) does not have a breastfeeding room, but they asked me to use the staff lounge, but people came in and out." *Violet, Malaysia.*

"They asked me to pump inside the toilet and I feel that is not good for myself and the breastmilk also, so I just slowly stopped" *Adeline, Malaysia.*

Responses to the COVID-19 pandemic provided more examples of how breastfeeding is not a priority for policy makers. During the lockdown period, participants described how visiting hours at hospitals were very limited and how policies included mothers of newborns admitted to the neonatal units. Lactation consultants were not considered 'essential medical

services' and there was limited access to them during the pandemic. Many participants reported that even after the strict lockdown was over and mothers and babies were allowed out, chairs in public places were still crossed out, and thus mothers had nowhere to sit and breastfeed.

> "I requested that I want to go to the hospital to direct-latch him, and they actually allowed me to do that. . .of course, only during their visiting time during this COVID situation (which was twice a day)" *Terri, Malaysia.*

> "Because of pandemic restrictions, my condo (apartment) doesn't allow any visitors" *Violet, Malaysia.*

> "And myself I didn't experience this but in [name of city] a lot of women who were a breastfeeding during the lockdown they go to the mall or something and they have crossed out all the chairs, so you couldn't you didn't have spaces to sit down to breastfeed" *Isabelle, Ireland*

## Major Theme 2: When breastfeeding is visible in the environment, it becomes normalised

This theme describes how visibility of breastfeeding within a community plays a role in normalizing breastfeeding, which perceived as an important measure of breastfeeding-friendliness. The mothers also brought up breastfeeding in public and expressed different views about it. This theme can be divided into four subthemes: effects of breastfeeding visibility, visuals of breastfeeding as part of everyday life, breastfeeding in public and public breastfeeding rooms are more for the babies.

**Effects of breastfeeding visibility.**　One of the main effects of breastfeeding visibility is that it creates a safe and welcoming space for breastfeeding mothers. The mothers expressed this in different ways, including how seeing others breastfeeding would make them not feel out of place.

> "Just seeing people around (breastfeeding).., that's the big thing, just like that you could somehow feel that no one's going to say anything to you. . ... kind of made you feel a bit safer" *Jade, Ireland*

> "I think that's the biggest support really would be to make it more common, to seeing people. . .I think it has to be like that, it becomes more common and you don't feel out of place" *Eva, Ireland.*

Another positive effect of breastfeeding visibility was the impact on one's decision to breastfeed. Having seen another person breastfeed, especially someone they know, helped some women decide to breastfeed their own children, even if the encounter was many years before they had children. Additionally, one mother described how when those around her saw her breastfeeding, they became more positive about the behaviour.

> "I saw my eldest sister breastfeeding and I also saw a very close friend of mine breastfeeding. . .it was like "wah!" It was something that really imprinted in my mind that it was the best thing to give to her baby" *Elise, Malaysia*

> "My mother, I think, she didn't even like the idea of someone breastfeeding. . .but after me being around and just feeding the baby on demand, she could see the positive aspects of breastfeeding, and was very positive towards breastfeeding after that." *Kristen, Ireland*

A strong visual message had a similar effect as seeing other breastfeeding women around. One mother saw a poster as a teenager in a public place, and it made such an impact on her that she could recall it when the time came for her to make a decision about feeding her baby.

"I went there for a music camp and then there was a poster with a lady feeding twins. Ever since then it is my head. . . And I didn't think that 20 years later when I was pregnant, I decided, yes, I want to breastfeed." *Camelia. Ireland.*

**Visuals of breastfeeding as part of everyday life.** Making breastfeeding visible does not have to mean women breastfeeding in public all the time. To the mothers interviewed, a big part of this was seeing the visuals of breastfeeding incorporated into everyday life encounters. This includes portraying breastfeeding as the normal way of feeding a baby in all forms of media. Even small gestures, like not using a bottle to indicate a baby room also helps to normalize breastfeeding.

"It's just having things around like in books and children's books, when you see pictures of breastfeeding mothers or even cartoons, dramas and soap - just where it's not made a big deal" *Jade, Ireland.*

"Definitely advertisements on TV, I mean, we see so many advertisements on formula milk. . .create awareness posters, billboards where it promotes breastfeeding, instead of the importance of formula" S*tella, Malaysia*

"The baby feeding room symbol as a bottle and not a breast. It's just these kind of little things that are subconscious signifiers of what is considered normal" *Amy, Ireland*

**Breastfeeding in public.** There were divided views on what breastfeeding in public meant to the mothers, in particular, whether a breastfeeding room was necessary. Some believed that it is not proper to breastfeed in public, yet others said they would not hesitate to do so and there were those in between, where they could breastfeed without a room, but use a nursing cover or choose a discreet place. These differences seemed to be related to the individual's confidence, rather than cultural background. It is often believed that Asian women were more conservative and would approach breastfeeding in public differently but this was not apparent in this study. One mother from Malaysia was particularly confident about breastfeeding in public and said that she would 'stare back' if someone was looking at her breastfeeding.

"Breastfeeding should not be in front of everyone" *Tara, Malaysia.*

"And I've never been kind of shy or tried to hide away, because I think it's really important to try and normalize breastfeeding as much as possible, I think it's encouraging when you see other people out and about doing it." *Lilian, Ireland*

"If someone wants to see, it's fine, I'm just feeding my child" *Stella, Malaysia*

**Public breastfeeding rooms are more for the babies.** Although a breastfeeding room was one of the main items on the 'wish list' of a breastfeeding-friendly city, the reason for needing the room appeared to be more for the comfort of the baby. The room, as opposed to

breastfeeding in an open space, provides a quiet and comfortable environment where baby can feed without distractions.

> "(Babies) don't like to be covered up and they need to have a proper room and get seated properly, then, only they can nurse well, nurse properly and all. So, baby rooms, that would be better." *Violet, Malaysia*

> "And it's not that I'm embarrassed to feed in a restaurant. I have no problem doing that but as children get older, they're more distracted." *Grace, Ireland*

Where there are no rooms, mothers would seek for alternatives of this type of space. An interesting alternative space was their car. It was not the most comfortable place, but several mothers described how it gave them the privacy needed.

> "So (we) try our best to comfort them and then rush to the car and latch them" *Adeline, Malaysia, mother of twins.*

> "Like I just did it (expressing milk) in back of my car every time that I was out and which not lovely, but it's fine" *Tessa, Ireland*

## Major Theme 3: Seamless and continuous breastfeeding support for everyone

This theme directly addressed the warm chain of breastfeeding. Seamless breastfeeding support is essential throughout the continuum of care and all level of society. The women highlighted gaps in support, and these were viewed negatively. These gaps can be seen in 3 main areas, corresponding to the subthemes: continuity of support across sectors, closing the gaps within the sector, and equity of access to breastfeeding support. Sectors refer to the three main sectors in the warm chain (Fig 1).

**Discontinuity of support across sectors.** One of the obvious issues identified from the interviews was that breastfeeding supports are often in-silo, not taking account the fluidity of movement of the breastfeeding dyad across the continuum of the warm chain. For example, the mothers noted that the positive support they received while at the hospital came to an abrupt stop once the new mother and baby went home. The transition from hospital to home is not easy for all women and for some, the connection to the community support group was not readily known to them and some weren't provided information about breastfeeding support groups in the community while they were in the hospital. Similarly, transitioning from home to work was described as challenging and they could not always find the support they needed. Again, this highlights that having a seamless support system helping mothers to transition from one sector to another is important.

> "So, once I went home, we just figure it out yourself but while they were there (i.e. at the hospital), they were helpful" *Camelia, Ireland.*

> "I think it would be a nice thing if the people from those groups could drop by their hospital once in a while to meet the people who are there" *Isabelle, Ireland.*

> "I find there was very little support for extended breastfeeding. I went back to (support group) breastfeeding meeting at around nine months before I went back to work, looking for advice, but everybody there had tiny babies and nobody who was still breastfeeding was still attending the classes." *Kristen, Ireland*

**Closing the gaps within the sector.** Within each sector, there were gaps in the quality and consistency of the support provided for the mothers. In the community, while there were support groups and access to lactation consultants, there was difficulty in accessing donor breast-milk. A mother from who could not find donor breast milk for her newborn described how there are "blood banks out there, but there are no breast milk banks."

In the healthcare sector, where breastfeeding support was reported as most constant, the quality of support given was inconsistent. For example, preterm babies were encouraged to be given expressed breast milk, but the mother was not taught or assessed if she knew how to express her milk. Health professionals, particularly general practitioners, lacked the expertise to detect problems related to breastfeeding, much to the disappointment of the mothers.

"They just passed me a syringe and then close the curtain. I just hide inside and hand express myself" *Adeline, Malaysia.*

"I saw four GPs about it, and none of them diagnosed it as a milk allergy. . .I just felt, that I was let down a little bit even in terms of being taken seriously about the concerns" *Amy, Ireland*

In the workplace, policy and practice seemed to be incongruent. At the time of the interviews, breastfeeding breaks were available nationally in Ireland until the baby was 26 weeks old [33]. There were some sectors (e.g. teachers, civil servants, and healthcare workers), who had worked through their unions to have breastfeeding breaks until their child was 104 weeks. However, utilization of these breaks can be challenging as described by the teachers in this study.

"A lot of secondary school teachers aren't taking up the breastfeeding breaks because it's too much of an imposition on their colleagues, and on their students" *Lilian, Ireland*

"So there wouldn't be anyone to supervise my class. I could have arranged and management were really supportive, but I just felt it wasn't worth my while to do that, this time, and also because my daughter was a bit older (over 1 year), I felt she was fine without it just for those few hours" *Grace, Ireland*

**Inequitable access to breastfeeding support.** Equity in access to skilled lactation support was described as an issue by some of the mothers. In Ireland, there were few lactation consultants in the public health system and many women would engage private lactation consultation services. In Malaysia, all lactation consultations are privately based. This means that many mothers had to pay for access to the lactation consultants. The extra cost would mean that some women were not able to afford this service.

"Yeah, it was just that was for me breastfeeding was really handy but I couldn't afford a lactation consultant because that's 50 euro - that goes towards food." *Nikki, Ireland*

Inequity in breastfeeding support was also apparent for mothers in the informal work sector. As described earlier, there was strong policy for workplace support in Ireland. In Malaysia, there is a no clear policy for breastfeeding other than mandatory maternity leave, but many formal workplaces would provide minimal support at the workplace such as a dedicated room and freedom to express breast milk at work. However, for women in the informal workforce,

these are not present and success or failure to continue breastfeeding depended on the individual mother's determination and resilience.

For example, Ruby from Malaysia works on-the-go, sometimes between 7am until 12 midnight. She would express breast milk in her car or at shopping malls while having her lunch. She described keeping her expressed breast milk in a cooler box in her car, occasionally having to add ice into the box midway through her day. In contrast, Tara also from Malaysia, who runs a small food business by the road does have any way of expressing breast milk while she is at work.

"I breastfeed him when I am back home but when I am out (working), he drinks formula milk from the bottle at home" *Tara, Malaysia.*

## Discussion

The ideal breastfeeding-friendly city was epitomized in the themes developed. Although the participants in our study came from two very different cultures, they generally wished that breastfeeding be prioritized and normalized and continuously supported in the place where they live.

One of the main characteristics of a breastfeeding friendly city described by mothers was a place to breastfeed their babies freely. While many would have preferred that breastfeeding could be normalized and thus no need to shy away from breastfeeding in public, this was idealistic rather than practical. Therefore, they had a pragmatic request for breastfeeding rooms to be available around the city. However, the women in our study had another reason for wanting a private room which is related to the baby. This is phenomenon is not reported in other studies about breastfeeding in public [34]. It is important to note that the general acceptance of public breastfeeding in Ireland has been reported as good [35], while in Malaysia, women has been reported to generally feel uncomfortable about breastfeeding in public [36, 37]. The experiences shared, and views expressed by the women in our study about breastfeeding while out and about resonated with other studies. Sheehan *et al.* explored the views of first-time expectant mothers and their families' views on breastfeeding in public [38]. This Australian study found that not everyone can accept breastfeeding in public and there's a general view that some discretion is needed. In that study, the car was also named as one such space to breastfeed away from the public view. Considering the point of view of the public, the acceptance of public breastfeeding is variable. A study by Morris *et al* found that most netizens in the United Kingdom do not immediately welcome this [39]. Therefore, creating accessible and welcoming public spaces for women to breastfeed is needed in a breastfeeding-friendly city, whether this includes a public breastfeeding room or not.

It is clear from epidemiological evidence that where there is high priority for and concentration of breastfeeding support, there will be higher breastfeeding rates. For example, the breastfeeding initiation rate before hospital discharge (at birth and within one day of birth) in Malaysia was 89.6% in 2016, but by six months, the prevalence of exclusive breastfeeding dropped to 47.1% [28]. Similar trend is seen in Ireland, with 63.1% of babies breastfed at birth in 2021, but by three months, 31.1% were exclusively breastfed [27]. Evidence from other studies looking at intention to breastfeed and actual breastfeeding duration also demonstrated that a majority of the mothers started well but stopped breastfeeding early [6]. Hence, the importance of having sustained and continuous support for breastfeeding beyond the perinatal period. Having a city that support breastfeeding across all time points in the breastfeeding journey could improve these outcomes.

The question arises if cities are currently built to support breastfeeding. An ethnographic study of a city in Wales, United Kingdom found that the city centre and its public transport areas were not the most 'friendly' places for women to breastfeed [40]. The study reported a busy and small waiting room at the railway station with chairs facing each other, making it difficult for a mother to "discreetly" breastfeed. It is interesting to note that the researcher found that while cafés are comfortable and private enough to breastfeed, the rapid pace of change of customers in a particular café could potentially mean that mothers sitting down to breastfeed their babies for up to twenty minutes may be frowned upon by the staff. Therefore, providing the infrastructure alone would not be enough and a culture change of accepting that breastfeeding is what babies do normally is also needed.

Many of the women in our study suggested that one way to normalize breastfeeding is to have pictures of breastfeeding as posters in public spaces or incorporated into everyday life such as books and television. An analysis of images of breastfeeding by Giles offers a good insight into what cities can do when using pictures and images of breastfeeding [41]. A picture tells a thousand words; therefore, the choice of image portraying breastfeeding is important. Many images show breastfeeding women alone, with the absence of other people, especially non-breastfeeding individuals. According to Giles, this portrays breastfeeding as something private, even though the pictures are publicly displayed. To change culture on breastfeeding, images of breastfeeding women engaging socially with non-breastfeeding community members could be the way forward. Some examples would be the works of a Malaysian artist, Chuah Thean Teng, who often depicts a breastfeeding mother in daily life activities [42], as well as Irish artist Fiona Carey with her collection on "Everyday Breastfeeding" (https://www.fionacarey.com/illustration).

A supportive and enabling environment is essential for breastfeeding success, no matter the geographic location. However, the specific needs may differ from place to place depending on the social, economic and cultural setting. This study exemplified the difference between a high-income and middle-income country from two different continents. In the context of workplace support, there was a clear difference. Women in Ireland were privileged to a much longer maternity leave (at least six months) and breastfeeding breaks. In contrast, Malaysian women had only two months maternity leave (note: a new legislation after the interview provides 90 days of paid maternity leave) and no breastfeeding breaks. Data from the International Labour Office showed that in 2022, 67.8% of Irish women received maternity benefits compared to 46.5% of Malaysian women (data from 2020) [43].

Another difference is the education levels of the women from both countries. There was a significantly higher number of women with a higher degree in Ireland compared to Malaysia. This is not likely to have occurred by chance because Ireland ranks as one of the top 10 countries with the highest number of PhD holders [44]. The difference in education level could have contributed to how much the women 'tolerated' any lack of support and what they did to overcome that. Women who are not cognisant of what they could have received in terms of support would generally accept what was available and would not seek out further support. From our sample, women with higher education or those who had more prior knowledge of breastfeeding support seemed to seek out for themselves support that was missing.

Although the specific examples highlighted the differences, the main themes remain similar regardless of the location. Even with the obvious socioeconomic and cultural differences between the two countries, breastfeeding women had similar perspectives when it came to the types of support they require. Other international studies such as our study also demonstrated that the support needs of breastfeeding women are universal [45, 46].

A key observation from the interviews were that almost all of the women demonstrated an extraordinary determination to breastfeed, despite facing obstacles and challenges. They went

to great lengths to make breastfeeding work for them. Some even had to 'fight' for their right to breastfeed, when told by healthcare professionals that they should introduce formula. The strong determination fits the description of the 'passionate' group of women as reported by another study in Malaysia [47]. However, not all breastfeeding women will fall into the 'passionate' group. Success in breastfeeding should not be dependent on individual motivation. The original breastfeeding friendly statement, the Innocenti Declaration, called for efforts to increase the individual mother's confidence and stated that "obstacles to breastfeeding within the health system, the workplace and the community must be eliminated" [48]. From the interviews in this study, there were many instances of obstacles to breastfeeding still present even though it has been over 30 years since the declaration. An enabling environment would negate the need for the individual mother to be incredibly passionate or motivated to succeed in breastfeeding.

At the time of the interviews, restrictions related to Covid 19 were still present. Many of the mothers had their babies during the peak of the pandemic. The experiences of the women during that time demonstrated how an emergency situation amplified existing inadequacies of breastfeeding support. At that time, breastfeeding was not considered a priority or essential healthcare. Incidences shared by some women such as restricted visiting hours, and not allowing lactation consultants to enter apartment building because it was considered non-essential, were some examples. Part of the issue was the slow understanding of the impact of the virus on breastfeeding [49]. By the time policies changed, many babies would have missed the opportunity to breastfeed, some exclusively and some altogether. A cross-sectional survey of post-partum women in 5 countries found that they experienced poor breastfeeding support during the pandemic [50]. Breastfeeding during other emergencies such as natural disasters and war also faced similar challenges [51, 52]. One issue observed from these situations is the uncontrolled distribution of breastmilk substitutes taking precedence over helping mothers to overcome difficulties in breastfeeding [53]. This again highlights the fact that breastfeeding needs to be at the front and centre where mothers and babies are concerned.

## Strengths and limitations

The methods used in this study accounted for potential cultural differences between the two countries. Our cross-cultural research team had either worked or are currently working in one of the countries. We also ensured that the interviews would be comprehensible to all participants by piloting the interviews with women from both countries. Consistency was maintained by having a single researcher conduct all interviews. The themes were developed specifically by two authors, one each from Ireland and Malaysia. In presenting the results, additional background information and the contextual nuances of the quotes were added for a deeper understanding of our findings.

The primary limitation of this study is that we interviewed only women who are currently breastfeeding or had breastfed before. This may present a biased view as they were more likely to be those who are highly motivated to breastfeed. However, among them, they had their fair share of positive and negative experiences. The unsuccessful experiences with breastfeeding gave them insights into the type of breastfeeding support needed. In addition, given that women were recruited from two large breastfeeding advocate groups, there is a potential for over-representation of like-minded viewpoints.

## Conclusions

The views and experiences of breastfeeding women across two continents demonstrated the importance of prioritized, and continuous support throughout the breastfeeding journey.

Their hopes and aspirations of a breastfeeding-friendly city would be useful for cities to consider when developing or implementing breastfeeding support programmes. The findings can also be used to guide development of indicators of a breastfeeding-friendly city.

## Acknowledgments

Ting Kai Xin for translation of the Malay interviews.

Page administrators of the two Facebook groups we used to recruit the participants.

All the women who responded to our call for participants and those who took time for the interviews.

## Author Contributions

**Conceptualization:** May Loong Tan, Elizabeth J. O'Sullivan, Jacqueline J. Ho, Amal Omer-Salim, Fionnuala M. McAuliffe.

**Data curation:** May Loong Tan.

**Formal analysis:** May Loong Tan, Elizabeth J. O'Sullivan.

**Investigation:** May Loong Tan.

**Methodology:** May Loong Tan, Elizabeth J. O'Sullivan, Jacqueline J. Ho, Amal Omer-Salim, Fionnuala M. McAuliffe.

**Supervision:** Jacqueline J. Ho, Fionnuala M. McAuliffe.

**Writing – original draft:** May Loong Tan, Elizabeth J. O'Sullivan.

**Writing – review & editing:** May Loong Tan, Elizabeth J. O'Sullivan, Jacqueline J. Ho, Amal Omer-Salim, Fionnuala M. McAuliffe.

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
