## [Decision Letter · Decision Letter 0]

27 Aug 2024

PONE-D-24-22876What makes a city breastfeeding friendly? A qualitative analysis of interviews with breastfeeding women from Europe and AsiaPLOS ONE

Dear Dr. Tan,

Thank you for submitting your manuscript to PLOS ONE. After careful consideration, we feel that it has merit but does not fully meet PLOS ONE’s publication criteria as it currently stands. Therefore, we invite you to submit a revised version of the manuscript that addresses the points raised during the review process.

- Ensure that your abstract adheres to our Instructions for Authors' formatting guidelines.

- Kindly include the requested supplementary file.

-There is a concern about the use of participants' names in the text. Please clarify whether the consent form explicitly allowed for the publication of their names in the study. Ensuring that participants were fully informed about this aspect of the publication process is crucial for maintaining ethical standards.

-Clarification is needed on how interview participants were selected from the larger pool of individuals who filled out the online forms. Please provide details on the criteria or process used to choose these participants.

-While the paper mentions obtaining institutional ethical approval, it would be beneficial to specify which ethical guidelines or frameworks were followed (e.g., Declaration of Helsinki, Belmont Report) to enhance the study's ethical transparency.

We look forward to receiving your revised manuscript.

Kind regards,

Trhas Tadesse Berhe, PhD

Academic Editor

PLOS ONE

Journal Requirements:

1. When submitting your revision, we need you to address these additional requirements. Please ensure that your manuscript meets PLOS ONE's style requirements, including those for file naming. The PLOS ONE style templates can be found at https://journals.plos.org/plosone/s/file?id=wjVg/PLOSOne_formatting_sample_main_body.pdf and https://journals.plos.org/plosone/s/file?id=ba62/PLOSOne_formatting_sample_title_authors_affiliations.pdf 2. In the online submission form, you indicated that "Data cannot be shared publicly because of confidentiality. De-identified data may be made available upon reasonable request from the corresponding author." All PLOS journals now require all data underlying the findings described in their manuscript to be freely available to other researchers, either 1. In a public repository, 2. Within the manuscript itself, or 3. Uploaded as supplementary information.This policy applies to all data except where public deposition would breach compliance with the protocol approved by your research ethics board. If your data cannot be made publicly available for ethical or legal reasons (e.g., public availability would compromise patient privacy), please explain your reasons on resubmission and your exemption request will be escalated for approval. 3. Please include your full ethics statement in the ‘Methods’ section of your manuscript file. In your statement, please include the full name of the IRB or ethics committee who approved or waived your study, as well as whether or not you obtained informed written or verbal consent. If consent was waived for your study, please include this information in your statement as well.

Reviewers' comments:

Reviewer's Responses to Questions

**Comments to the Author**

1. Is the manuscript technically sound, and do the data support the conclusions?

Reviewer #1: Yes

Reviewer #2: Yes

2. Has the statistical analysis been performed appropriately and rigorously? 

Reviewer #1: Yes

Reviewer #2: Yes

3. Have the authors made all data underlying the findings in their manuscript fully available?

Reviewer #1: Yes

Reviewer #2: No

4. Is the manuscript presented in an intelligible fashion and written in standard English?

Reviewer #1: Yes

Reviewer #2: Yes

5. Review Comments to the Author

Reviewer #1: The manuscript is well-written overall. However, please review the attached documents and address all the comments and suggestions provided. This will help enhance the quality and clarity of your paper.

Reviewer #2: Thank you for requesting me to review the paper “What makes a city breastfeeding friendly? A qualitative analysis of interviews with breastfeeding women from Europe and Asia”.

General comments

This is a great work and well-written. There are various points where participant’s names were mentioned. I want to inquire whether the consent form specified that their name should appear in the publication.

Other comments:

1. Lines 143 – 149, when was the demographic characteristics data collected (Dates started and ended)?

2. Methods: Were the interviews conducted during the period for demographic characteristics data collection? How were the participants for interviews chosen from the many participants who filled in the online forms?

3. Line 189 – 191 can be transferred to the methods section.

4. Selection criteria, line 192 can also be taken to the methods section.

5. The discussion was generally well done.

Thank you,

6. PLOS authors have the option to publish the peer review history of their article (what does this mean?). If published, this will include your full peer review and any attached files.

Reviewer #1: **Yes: **Seid Muhumed Abdilaahi (Departments of Pediatrics and Child Health Nursing, Institute of Health Science, Jigjiga University, Jigjiga, Ethiopia)

Reviewer #2: **Yes: **Silvia Awor

---

## [Author Response · Author response to Decision Letter 0]

5 Dec 2024

Here are our responses (in blue) to the comments, feedback and questions:

1. Ensure that your abstract adheres to our Instructions for Authors' formatting guidelines:

We have checked that our abstract adheres to the instructions as per https://journals.plos.org/plosone/s/submission-guidelines.#loc-abstract

2. Kindly include the requested supplementary file:

 We are unable to include the requested supplementary file because our data comprises of transcripts of the interviews. The transcripts may potentially identify the participants or institutions/hospitals/organizations even if all identifying details are removed. Therefore, sharing this openly would breech confidentiality. We have revised our statement on data sharing to: “Data cannot be shared publicly because of confidentiality and restrictions from research ethics committees as the data in this paper comprises confident transcripts of interviews. De-identified data may be made available upon reasonable request from the corresponding author."

3. There is a concern about the use of participants' names in the text. Please clarify whether the consent form explicitly allowed for the publication of their names in the study. Ensuring that participants were fully informed about this aspect of the publication process is crucial for maintaining ethical standards: 

We would like to clarify that the names in the manuscript are pseudonyms. We have added this in the text to make this clear (Lines 196-197). Participants were fully aware that we will use pseudonyms when quoting them in any publications and consented to this in the consent form. The identities of all participants are kept confidential according to ethical standards.

4. Clarification is needed on how interview participants were selected from the larger pool of individuals who filled out the online forms. Please provide details on the criteria or process used to choose these participants: 

Additional information is added to the methods section to indicate how we selected the participants form the larger pool (Lines 160-167). In general, it was based on our predetermined criteria as well as availability of the participants. The interviews stopped when data saturation was reached.

“Selection of participants for interview:

All the women who completed the online and screening form, and met the inclusion criteria, were contacted for interviews. Initially, the women were contacted based on the timing of receiving the forms. They were contacted within one week of receiving the screening form, and if they responded, interviews scheduled within two weeks. If a participant did not respond to our contact for an interview appointment, the next person with similar characteristics was selected. As the number of participants grew, preference was then given to those with characteristics not yet met in the earlier participants. This continued until information saturation was met.”

5. While the paper mentions obtaining institutional ethical approval, it would be beneficial to specify which ethical guidelines or frameworks were followed (e.g., Declaration of Helsinki, Belmont Report) to enhance the study's ethical transparency: 

We have edited it to specify that the study followed the framework of Declaration of Helsinki (Lines 111-116).

“The study was conducted following the principles of the Declaration of Helsinki and received approval from Joint Penang Independent Ethics Committee (No. JPEC 21- 0011, 4 July 2022), as well as declared as review exempt from UCD Human Research Ethics Committee (No: LS-E-22-139-Tan-McAuliffe, 14 July 2022). Electronic written consent was obtained before the commencement of the interviews, which included consent for the recording of the interviews and use of pseudonyms when quoting them”

6. Response to Reviewers’ Specific Comments:

Reviewer 1

Comment Authors’ Response Reference in the manuscript

The manuscript is well-written overall. However, please review the attached documents and address all the comments and suggestions provided. This will help enhance the quality and clarity of your paper Thank you.

Please see the attached file for responses to the comments found in the document.

Each of the comments are also addressed below: 

Abstract: 

The term "warm chain of support" is well-defined, which helps set the context. However, the definition is quite dense and might benefit from being broken down into simpler components or rephrased to ensure it's easily understood by a broad audience. Thank you. We are unsure about this comment if it’s in general or specific to the abstract. We note the other comment on expanding the definition in the background. We have taken that suggestion in the background but have decided to leave this definition as it is in the abstract due to word limit. 

The methods section is concise and describes the sampling technique, participant demographics, and analytical framework used. However, the explanation of "Purposive sampling" could be simplified or elaborated on slightly to ensure all readers understand its significance.

 Thank you.

We have replaced the term purposive sampling in the abstract to “A set of selection criteria was applied to….”

 Line 22

Introduction:

Explicitly Address the Research Gap: While you allude to the importance of an enabling environment, directly stating the lack of standardized indicators for breastfeeding-friendly cities would significantly enhance the introduction's impact. Currently, the text implicitly assumes that readers understand this gap in the literature. Consider adding a sentence like, "Despite the recognized need for supportive environments, there is currently a lack of consensus on what constitutes a breastfeeding-friendly city, hindering the development of effective interventions and policies." This addition would clearly articulate the need for your research and its potential contribution to addressing this gap.

Seamlessly Introduce the Study's Purpose: While the abstract states the study's objective, briefly mentioning the purpose within the introduction would create a smoother transition to the subsequent sections and provide readers with a clearer understanding of the research direction. For instance, you could incorporate a sentence like, "This study aims to explore the perspectives of breastfeeding women in Ireland and Malaysia to understand the key elements of an ideal breastfeeding-friendly city." This addition would link the broader context of breastfeeding support to your specific research aims.

 Thank you. We have added the suggested sentence:

"Despite the recognized need for supportive environments, there is currently a lack of consensus on what constitutes a breastfeeding-friendly city, hindering the development of effective interventions and policies."

The subsequent sentence was modified so that the thought flows better. 

With regards to the aims and objectives, we agree that it wasn’t clear, we have taken your suggestion and changed it to: 

“This study aims to explore the perspectives of breastfeeding women from two continents – Europe and Asia, to understand the key elements of an ideal breastfeeding-friendly city”

In addition, to highlight the study’s purpose, we moved it to a new paragraph. Lines 85-88 

Lines 98-99

Line 98

The warm chain concept is crucial to the study, yet the explanation is slightly dense. Consider adding a brief, clear definition or example of how the warm chain operates in practice, which could help readers unfamiliar with the concept to better grasp its significance.

 We added a sentence to make the definition clearer: “The warm chain is akin to the cold chain in vaccine delivery whereby a break within the chain would compromise the outcome, in this case, the duration of breastfeeding.” 

We also drew attention to the readers to Figure 1 to visualize what it means.

“See Fig 1 for a diagrammatic representation of the warm chain.”

 Line 59-61

Responses to the editor and reviewers are attached as a separate file.

Line 62-63

The shift towards Public-Patient Involvement is noted, but the importance of this shift could be emphasized more. A brief statement on how incorporating women’s perspectives directly influences the effectiveness of breastfeeding support programs could strengthen the rationale for your study.

 Thank you for the suggestion. We have added these to strengthen our rationale:

“Policies and interventions developed without input from end-users may fail to gain widespread adoption, as they may not adequately address their needs. Similarly, a breastfeeding-friendly city must be designed with the active participation of breastfeeding women.” Lines 94-96

Clarify the difference between "perceived" and "perceive." We used the term ‘perceived’ as in the past tense of perceive. It is in the past tense because the perception is based on past experiences.

The sentence has now been replaced by Line 98-99, which does not have the word “perceived”. 

While the you have mentioned obtaining institutional ethical approval, it does not specify which ethical guidelines were followed. Including this information would enhance the ethical transparency of the study.

 The manuscript had the details of the ethical approval on a separate page (not part of the main manuscript). 

We now added details on the ethical approval in the main manuscript:

“The study was conducted following the principles of the Declaration of Helsinki and received approval from Joint Penang Independent Ethics Committee (No. JPEC 21- 0011, 4 July 2022), as well as declared as review exempt from UCD Human Research Ethics Committee (No: LS-E-22-139-Tan-McAuliffe, 14 July 2022). Electronic written consent was obtained before the commencement of the interviews, which included consent for the recording of the interviews and use of pseudonyms when quoting them” 

Lines 111-116

Although you’ve described the use of purposive sampling and personal contacts to recruit participants. However, there is no discussion of potential biases that might arise from this method, such as selection bias or the influence of the researchers’ networks on the sample diversity. Addressing these concerns would add rigor to the sampling methodology.

 Thank you for highlighting this. To note, the researchers’ networks are rather diverse because of the experience of the main researchers in the field of breastfeeding within each country (Ireland and Malaysia)

We have added to our discussion on limitation the potential selection bias – the women were recruited from among those who are breastfeeding and are likely advocates for breastfeeding because the belong to a similar group:

“In addition, given that women were recruited from two large breastfeeding advocate groups, there is a potential for over-representation of like-minded viewpoints.” 

Lines 625-627

The interviews:

Single interviewer conducted all the interviews. While this ensures consistency, it could also introduce interviewer bias.

 Interviewer bias is unlikely to impact or influence the findings of our study. The study explored both the positive and negative aspects of breastfeeding support as experienced by the women. The structured questions also ensured that both sides of the story are being heard. 

The explaination of the coding process and mentions that the lead investigators generated and reviewed the themes. However, it does not address how potential biases from the researchers’ backgrounds and experiences were managed during the analysis. Discussing any steps taken to ensure objectivity, such as peer debriefing or involving an external auditor, would enhance the credibility of the findings.

 The researchers’ background and experiences add to the reflexivity of the process. The analyses were completed from our personal lenses, which is the principle of reflexive thematic analysis by Braun & Clarke. We did not include external auditors. However, the themes were presented to the other authors for discussion and all authors, not only the investigators who developed the initial themes, agreed to the findings.

We added the following to make it clear: “The themes were presented to the remaining authors (JJH, AOS and FMM) to ensure objectivity and address any bias that may have occurred”. 

Lines 207-208

Results:

The statement that "many of the women would have had their babies born during the full lock-down period" is speculative. Consider providing specific data or revising the language to clarify whether this was confirmed for the participants or is an assumption.

 This was a fact confirmed by the participants and not an assumption.

We have revised the sentence to:

“During this period, both countries were still in partial lockdown due to the COVID-19 pandemic and many of the women had their babies born during the full lock-down period of the pandemic.” 

Lines 212-213

You have states that 40 women expressed interest, but only 22 were interviewed. It would be beneficial to explain why the other 18 were not included (e.g., did not meet inclusion criteria, declined, or were not needed after saturation was reached).

 We reviewed our dataset, and confirmed that it was 42 (not 40) who expressed interest. The 20 women who were not included either because they could not be reached to arrange for interviews or we had reached saturation, and thus not contacted (4 declined, and we didn’t need to contact 16) 

Therefore, we have revised the sentence to:

“We interviewed a total of 22 women from a variety of ethnic/cultural backgrounds living in Malaysia (n=11) and Ireland (n=11). The equal number in the two countries was coincidental. Four women did not respond to our invitation for an interview and 16 were not needed after selection based on the demographic characteristics and subsequent information saturation.”

We also added a section describing the selection process with we hope help make this clear. 

“Selection of participants for interview:

All the women who completed the online and screening form, and met the inclusion criteria, were contacted for interviews. Initially, the women were contacted based on the timing of receiving the forms. They were contacted within one week of receiving the screening form, and if they responded, interviews scheduled within two weeks. If a participant did not respond to our contact for an interview appointment, the next person with similar characteristics was selected. As the number of participants grew, preference was then given to those with characteristics not yet met in the earlier participants. This continued until information saturation was met.” 

Lines 214-218

Lines 160-167

You indicate that an equal number of participants were interviewed from Malaysia and Ireland (n=11 each). Consider discussing whether this equal representation was intentional and if so, why it was important for the study’s objectives. If it was coincidental, acknowledging this might be useful. It was not intentional.

We added a sentence to acknowledge this.

“The equal number in the two countries was coincidental.” 

Line 216

You notes that participants from Ireland were generally older and more educated compared to those from Malaysia. This demographic difference could have influenced the findings. It would be helpful to mention whether these differences were considered in the analysis or interpretation of the results.

 We agree that the demographic differences influenced the findings. We did compare the findings between Ireland and Malaysia and these were discussed in the throughout the Discussion. Lines 563-580 specifically discussed the differences in demographic between the 2 countries. 

Table 1: Kindly move this in the last coloumn make the frist coloumn in Ireland followed by Malaysia in all tables.

 Thank you. We have taken up your suggestion and edited the table. Table 1

Table 1:

In this row, clarify that "Twins" and "Preterm delivery" are not mutually exclusive. Consider adding a footnote or combining these into a single count if some babies had both complications.

 There is a footnote to explain this. We prefer not to combine to show the difference.

---

## [Decision Letter · Decision Letter 1]

27 Dec 2024

What makes a city breastfeeding friendly? A qualitative analysis of interviews with breastfeeding women from Europe and Asia

PONE-D-24-22876R1

Dear Dr. May Loong Tan,

We’re pleased to inform you that your manuscript has been judged scientifically suitable for publication and will be formally accepted for publication once it meets all outstanding technical requirements.

Kind regards,

Trhas Tadesse Berhe, PhD

Academic Editor

PLOS ONE

Additional Editor Comments (optional):

Reviewers' comments:

Reviewer's Responses to Questions

**Comments to the Author**

1. If the authors have adequately addressed your comments raised in a previous round of review and you feel that this manuscript is now acceptable for publication, you may indicate that here to bypass the “Comments to the Author” section, enter your conflict of interest statement in the “Confidential to Editor” section, and submit your "Accept" recommendation.

Reviewer #1: All comments have been addressed

2. Is the manuscript technically sound, and do the data support the conclusions?

Reviewer #1: Yes

3. Has the statistical analysis been performed appropriately and rigorously? 

Reviewer #1: Yes

4. Have the authors made all data underlying the findings in their manuscript fully available?

Reviewer #1: Yes

5. Is the manuscript presented in an intelligible fashion and written in standard English?

Reviewer #1: Yes

6. Review Comments to the Author

Reviewer #1: (No Response)

7. PLOS authors have the option to publish the peer review history of their article (what does this mean?). If published, this will include your full peer review and any attached files.

Reviewer #1: **Yes: **Seid Muhumed Abdilaahi, Department of Pediatrics and Child Health Nursing, Institute of Health Science, Jigjiga University, Jigjiga, Ethiopia

---

## [Editor Report · Acceptance letter]

30 Dec 2024

PONE-D-24-22876R1 

PLOS ONE

Dear Dr. Tan, 

I'm pleased to inform you that your manuscript has been deemed suitable for publication in PLOS ONE. Congratulations! Your manuscript is now being handed over to our production team.

Kind regards, 

on behalf of

Dr. Trhas Tadesse Berhe 

Academic Editor

PLOS ONE